# Synthesis of Novel Aqua $\eta^4$-NNNO/Cu(II) Complexes as Rapid and Selective Oxidative Catalysts for O-Catechol: Fluorescence, Spectral, Chromotropism and Thermal Analyses

**Amjad M. Shraim** [1] , **Kifah S. M. Salih** [1] , **Ranim E. Al-Soufi** [1], **Soaad R. Al-Mhini** [1], **Mohammad I. Ahmad** [2] **and Ismail Warad** [1,*]

1   Department of Chemistry and Earth Sciences, College of Arts and Sciences, Qatar University, Doha P.O. Box 2713, Qatar; amjad@qu.edu.qa (A.M.S.); ksalih@qu.edu.qa (K.S.M.S.); ra1509392@student.qu.edu.qa (R.E.A.-S.); sa1303965@student.qu.edu.qa (S.R.A.-M.)

2   Central Laboratories Unit, Qatar University, Doha P.O. Box 2713, Qatar; mohammad.ibrahim@qu.edu.qa

*   Correspondence: ismail.warad@qu.edu.qa

**Abstract:** A new tetradentate Schiff base (SB), (E)-4-fluoro-2-(1-((2-(piperazin-1-yl)ethyl)imino)ethyl)phenol, was synthesized from condensation of 2-(1-piperazinyl)ethylamine and 5-fluoro-2-hydroxyacetophenone. This ligand was coordinated with three copper(II) salts ($CuCl_2$, $CuBr_2$ and $Cu(NO_3)_2 \cdot 3H_2O$) separately, giving rise to new neutral water-soluble Cu(II)/$\eta^4$-NNNO complexes (**1–3**). The new materials were fully characterized by standard spectroscopic, elemental, thermal, electronic, absorption, and fluorescence analyses. The chromotropism investigation of the aqueous solutions of the complexes revealed notable outcomes. A turn off-on halochromism effect was observed, both in the acidic and basic mediums. The green-colored solution was changed to colorless (off) upon the addition of HCl, while the initial green color was reversibly restored (on) after the addition of NaOH. On the other hand, bathochromic solvatochromism shifts were noticed in various solvents. Interestingly, complex **2** displayed a remarkable blue fluorescence shift ($\Delta\lambda = 90$ nm) when compared to its SB ligand. The oxidation capability of the three complexes was successfully demonstrated for the conversion of o-catechol to o-benzoquinone in aqueous solutions and in the presence of $H_2O_2$, an environmentally friendly oxidant, under mild reaction conditions.

**Keywords:** Cu(II) complexes; solvatochromism; halochromism; fluorescence; NNNOH Schiff base; catalytic oxidation

## 1. Introduction

The principle of hardness and softness of donor groups plays a vital role in coordination chemistry and is directly associated with the choice of the reaction conditions. Although this is considered as qualitative rather than quantitative description, it assists in understanding and determining the chemical properties and path of reactions [1]. Copper ion exhibits basic functions with different levels of activity in electron transfer, oxygen transport and reversible redox processes. Since Cu(II) is categorized as a species of intermediate Pearson acidity, the copper ion can coordinate with hard or soft binding functional group on ligands to deliver mono- or poly-nuclear complexes, possessing various coordination numbers ranging from 4 to 6 [2–4].

Development and implementation of coordination compounds bearing bidentate, tridentate or multidentate donor ligands attracted considerable interest during the last few decades for their chemistry and biological relevance. Schiff base (SB) or imine functional group [5] is a useful assembly extensively employed for constructing coordination compounds with a broad range of transition metals. The SB group, on the other hand, can stabilize diverse oxidation states of the metal center [6,7]. Copper(II) complexes with SB ligands have been used in a selection of significant applications including anticancer agents,

anticorrosive materials, optical materials and optical data storage [8–11]. Furthermore, such compounds act as catalysts in different organic transformations; for instance in Glaser–Hay acetylenic coupling, Baeyer–Villiger oxidation and various oxidation reactions of aromatic C-H, alkane, alkene, alcohol, catechol and sulfur-containing compounds [12,13].

The magnificent color of a wide spectrum of transition metal complexes results from different electronic transitions such as ligand to metal charge transfer (LMCT), metal to ligand charge transfer (MLCT) and the d–d electron transitions of the metal ions. Under physical or chemical variations, the color of a wide range of complexes is found to be reversible based on the ambient conditions. This phenomenon is well known as chemotropism, which is usually categorized based on the reaction or interaction type into solvatochromism, ionochromism, halochromism, electrochromism, thermochromism, piezochromism and photochromism [14,15].

In this context, we describe the synthesis of (*E*)-4-fluoro-2-(1-((2-(piperazin-1-yl)ethyl) imino)ethyl)phenol from condensation of 2-(1-piperazinyl)ethylamine and 5-fluoro-2-hydroxyacetophenone with complete characterization. As this ligand compound denotes an appropriate multi-dentate position via the O,N-donor atoms, the coordination with three copper(II) precursors in the absence of base was accomplished smoothly and in good yields. Exceptional chemotropism properties are expected to be demonstrated by the complexes under different conditions and physical states, due to the proton-transfer from the 2-hydroxyaryl group to the attached imine moiety on SB skeleton [16–21]. Hence, the chemotropism (solvent, ion, thermal), florescence and thermal behaviors of the SB ligand and the three new copper(II) complexes were examined. Furthermore, the catalytic oxidation of o-catechol to o-benzoquinone was achieved in an eco-friendly process employing the synthesized complexes in the presence of $H_2O_2$ as oxidant.

## 2. Experimental

### 2.1. Materials and Measurements

Solvent and fine chemicals were of reagent or analytical grade; all were directly put in use without any further purification. 5-fluoro-2-hydroxyacetophenone, 2-(1-piperazinyl) ethylamine and copper(II) bromide was acquired from Sigma-Aldrich. Silver nitrate, copper(II) chloride and copper(II) nitrate trihydrate were procured from Acros Organics. Bruker ALPHA FTIR spectrometer (Bruker GmbH, Berlin, Germany) was utilized to record transmission spectra in solid state in the range of 400–4000 cm$^{-1}$. Thermo Fisher Scientific™ FLASH™ 2000 Organic Elemental Analyzer was employed to evaluate the CHN analysis. JOEL™ 600 MHz NMR spectrometer (JOEL Ltd., Tokyo, Japan) was used to acquire $^1$H-, $^{13}$C- and $^{19}$F-NMR spectra with internal reference to the residual solvent signal, DMSO-$d_6$: δ = 2.50 ppm for $^1$H-NMR and δ = 39.52 ppm for $^{13}$C-NMR. Shimadzu GC-MS QP2010 and Shimadzu Spectro-fluorophotometer RF-6000 (Shimadzu, Tokyo, Japan) was used to obtain relative mass spectra and fluorescence, respectively. Agilent 8453 single beam spectrophotometer (Agilent, Santa Clara, CA, USA) was put to use to perform all UV-Vis measurements in methanol. NovaNano™ SEM 450 FEI, Netherlands-Energy Dispersive X-ray Bruker (SEM-EDX) Scanning Electron Microscope (Bruker GmbH, Berlin, Germany). System was used to acquire images and elemental composition. PerkinElmer TGA 4000 and PerkinElmer DSC 4000 were utilized to obtain the TGA and DSC analysis, respectively.

### 2.2. Synthesis

#### 2.2.1. Synthesis of SB Ligand

A solution of 584 mg (2.2 mmol) of 5-fluoro-2-hydroxybenzophenone in 10 mL of THF was added to a solution of 2-(1-piperazinyl)ethylamine (270 μL in 5 mL of THF (2.1 mmol)) at room temperature. The mixture was subjected to stirring for 20 h at room temperature until a yellow precipitate was formed. The precipitate was filtered and washed well with 20 mL of n-hexane and 40 mL of distilled water and then dried. The final product, (*E*)-4-fluoro-2-(1-((2-(piperazin-1-yl)ethyl)imino)-ethyl)phenol, was obtained in 80% yield, having a m.p. of 80–82 °C. Anal. Calcd. (%) for $C_{14}H_{20}FN_3O$: C, 63.37; H, 7.60; N, 15.84.

Found: C, 63.09; H, 7.83; N, 15.55. [M$^{+1}$] $m/z$ = 266.2 (265.33, theoretical). FT-IR: 3112 ($\nu_{C-H-Ar}$), 2910 ($\nu_{C-H}$), 1650 ($\nu_{C=N}$) cm$^{-1}$. UV-Vis absorbance in MeOH, $\lambda_{max}$ = 232, 329 and 404 nm. $^1$H-NMR (600 MHz, CDCl$_3$): δ 16.05 (s, 1H, OH), 7.33 (dd, *J* = 10.4, 3.2 Hz, 1H, Ph), 7.03–6.99 (m, 1H, Ph), 6.65 (dd, *J* = 10.4, 5.0 Hz, 1H, Ph), 3.54 (t, *J* = 6.6 Hz, 2H, C=NCH$_2$), 2.55 (br, 4H, 2 × CH$_2$-pipN), 2.46 (t, *J* = 6.6 Hz, 2H, CH$_2$N), 2.38 (m, 1H, N-H), 2.25 (br, 4H, 2 × CH$_2$-pip), 2.21 (s, 3H, CH$_3$) ppm. $^{13}$C-NMR (151 MHz) δ 171.78 (d, *J* = 3.2 Hz, C=N), 160.23, 153.38 (d, *J* = 231.2 Hz, C-F), 119.47 (d, *J* = 23.2 Hz), 119.20 (d, *J* = 7.5 Hz), 118.52 (d, *J* = 7.2 Hz), 113.98 (d, *J* = 23.7 Hz), 58.52, 54.28, 46.35, 45.59, 14.71 ppm.

### 2.2.2. Synthesis of Cu(II) Complexes: General Procedure

A solution of 3.60 mmol of each of the three copper(II) salts dissolved in 10 mL THF was poured into a solution of 3.60 mmol (1.00 g) of (*E*)-4-fluoro-2-(1-((2-(piperazin-1-yl)ethyl)imino)ethyl)phenol in 15 mL of warm THF with stirring. The color of the reaction mixture changed to green before a precipitate started to appear. Stirring was continued for 20 min at ambient temperature to ensure completeness of the reaction. The produced metal complex was filtrated off, washed with ethyl ether and n-hexane, and then dried under reduced pressure.

### Complex **1**

Upon mixing with the SB ligand, a mass of 0.48 g of CuCl$_2$ produced a green water-soluble powder of complex **1** with 64% yield, m.p. = 196 °C. Anal. Calcd. (%): C, 44.10; H, 5.55; N, 11.02 from C$_{14}$H$_{21}$ClCuFN$_3$O$_2$ molecular formula. Found (%): C, 43.85; H, 5.74; N, 11.19. FT-IR: 3450 ($\nu_{H2O}$), 3377 ($\nu_{H-N}$), 3126 ($\nu_{C-H-Ar}$), 2899 ($\nu_{C-H}$), 1621 ($\nu_{N=C}$), 1522, 1495 ($\nu_{NO2}$), 1442, 1384 ($\nu_{NO2}$), 1181 ($\nu_{N-C}$), 612, 540 and 452 ($\nu_{Cu-N}$). UV-Vis in water: $\lambda_{max}$ nm ($\varepsilon_{max}$ in M$^{-1}$ cm$^{-1}$): 255 (5 × 10$^4$), 375 (2.5 × 10$^4$), 670 (160).

### Complex **2**

Reacting a mass of 0.80 g of CuBr$_2$ with the SB ligand gave a green water-soluble powder of complex **2** with 67% yield, m.p. = 168 °C. Anal. Calcd. (%): C, 39.49; H, 4.97; N, 9.87 from C$_{14}$H$_{21}$BrCuFN$_3$O$_2$ molecular formula. Found (%): C, 39.40; H, 5.25; N, 10.03. FTIR: 3443 ($\nu_{H2O}$), 3352 ($\nu_{H-N}$), 3117 ($\nu_{C-H-Ph}$), 2869 ($\nu_{C-H}$), 1614 ($\nu_{N=C}$), 1530, 1494 ($\nu_{NO2}$), 1457, 1373 ($\nu_{NO2}$), 1173 ($\nu_{N-C}$), 614, 542 and 456 ($\nu_{Cu-N}$). UV-Vis in water: $\lambda_{max}$ nm ($\varepsilon_{max}$ in M$^{-1}$ cm$^{-1}$): 265 (3.0 × 10$^4$), 360 (2.2 × 10$^4$), 605 (120).

### Complex **3**

A mass of 0.87 g of Cu(NO$_3$)$_2$·3H$_2$O reacted with the SB ligand yielded dark-green water-soluble powder of complex **3** with 61% yield, m.p. = 250 °C. Anal. Calcd. (%): C, 41.23; H, 5.19; N, 13.74 from C$_{14}$H$_{21}$CuFN$_4$O$_5$ molecular formula. Found (%): C, 41.01; H, 5.46; N, 13.69%, FTIR: 3456 ($\nu_{H2O}$), 3351 ($\nu_{H-N}$), 3114 ($\nu_{C-H}$ of ph), 2879 ($\nu_{C-H}$), 1627 ($\nu_{N=C}$), 1522, 1493 ($\nu_{NO2}$), 1448, 1377 ($\nu_{NO2}$), 1163 ($\nu_{N-C}$), 614, 542 456 ($\nu_{Cu-N}$). UV-Vis in water: $\lambda_{max}$ nm ($\varepsilon_{max}$ in M$^{-1}$ cm$^{-1}$): 255 (1.0 × 10$^4$), 350 (7.2 × 10$^3$), 602(80).

### *2.3. Catalytic Oxidation of Catechol*

The oxidation of catechol was achieved using hydrogen peroxide in the presence of copper(II) complex as a catalyst. A mixture of 2.5 mmol of o-catechol, 5 mmol of 30% of hydrogen peroxide, 2.5 mol% of copper(II) catalyst and 10 mL of solvent (see Table 1) was heated at 50 °C for 5 min in a 20 mL glass tube. The reaction was monitored by UV-Vis focusing on the absorbance values at 358 nm of the 1,2-benzoquinone. Reaction mixture was transferred to a 50 mL separatory funnel and extracted twice with 10 mL of CH$_2$Cl$_2$. The organic layers were combined, dried, and evaporated to give a crude product of 1,2-benzoquinone to be confirmed by IR and NMR.

**Table 1.** Catalytic oxidation of benzyl alcohol.

| Entry | Complex | Solvent | Conversion %[a] | Time (min) | TOF[g] |
|-------|---------|---------|-----------------|------------|--------|
| 1 | **1** | Water | >99 | 10 | 625 |
| 2 | **2** | Water | >99 | 15 | 400 |
| 3 | **3** | Water | >99 | 20 | 303 |
| 4 | **1** | DMSO | >99 | 15 | 400 |
| 5 | **1** | DMF | >99 | 18 | 333 |
| 6 | **1**[b] | Water | >99 | 5 | 1250 |
| 7 | **1**[c] | Water | >99 | 2 | 3333 |
| 8 | **1**[b, d] | Water | 2 | 100 | 1 |
| 9 | **1**[b, e] | Water | 10 | 100 | 5 |
| 10 | **1**[b, f] | Water | 0 | 100 | 0 |

[a] GC-MS, [b] 50 °C, [c] 70 °C, [d] no catalyst, [e] no $H_2O_2$ and [f] acidic or basic medium, [g] turnover frequency.

## 3. Results and Discussion

### 3.1. Synthesis and Characterization

The SB derivatives are traditionally synthesized from the dehydration of amines with aldehydes or ketones under different reaction temperatures. The SB-functionalized ligand of interest, (*E*)-4-fluoro-2-(1-((2-(piperazin-1-yl)ethyl)imino)ethyl)phenol, was smoothly and quantitatively obtained from the condensation reaction of 5-fluoro-2-hydroxyacetophenone with 2-(1-piperazinyl)ethylamine within a day of stirring at room temperature. The prepared ligand is considered as an appropriate tetradentate ligand through one O and three N atoms accessible on the skeleton of the ligand structure. For these attractive chelating characteristics, three copper(II) complexes were independently synthesized from the corresponding copper(II) salts ($CuCl_2$, $CuBr_2$ and $Cu(NO_3)_2 \cdot 3H_2O$) and an equimolar ratio of ligand. In the absence of base, deprotonation/coordination reaction process of the tetradentate ligand allowed the coordination in an NNNO-style mode to the copper metal center (Scheme 1). For copper complexes similar to those reported in the current work, it has been suggested that the NNNO ligand does not coordinate via the piperazine NH part since it is already protonated by the H of the OH group [16–18]. The structure of these complexes has been proved by XRD [16]. Due to the ability of the Cu(II) center to form five-coordinate complexes, a stable trigonal bipyramidal geometry has been suggested as shown in Scheme 1. Moreover, Cu(II) center has been reported to make six bonds, where water as a moderate ligand plays a complementary role in forming six or five coordination numbers [18–25]. Notably, our ligand and its Cu(II) complexes showed high stability in addition to adequate solubility in typical polar solvents such as water, MeOH, DMSO and DMF.

**Scheme 1.** Synthesis of SB-functionalized ligand and it complexes.

All resonances for the predicted structure were confirmed by $^1$H-NMR spectroscopy in DMSO-$d_6$. A singlet peak at 2.21 ppm was attributed to the three protons of the methyl group. The N-H of piperazinyl ring was found as a multiplet peak at 3.54 ppm, whereas the two methylenes of piperazinyl moiety were noticed at 2.25 (br, $2 \times CH_2NH$) and 2.55 (br, $2 \times CH_2N$) ppm. Moreover, the protons of the two aliphatic methylene groups were detected as expected as triplets at 2.46 ($CH_2N$) and 3.54 ($C=NCH_2$) ppm. The three aromatic protons were observed at 6.65 (d), 6.99–7.03 (m) and 7.33 (s) ppm. Notably, a broad peak in the extremely downfield region, specifically at 16.05 ppm, was assigned to the phenolic proton. All the corresponding aliphatic, aromatic and iminic carbon signals were displayed in the $^{13}$C-NMR spectrum at the anticipated range of chemical shifts, as exemplified in Figure 1.

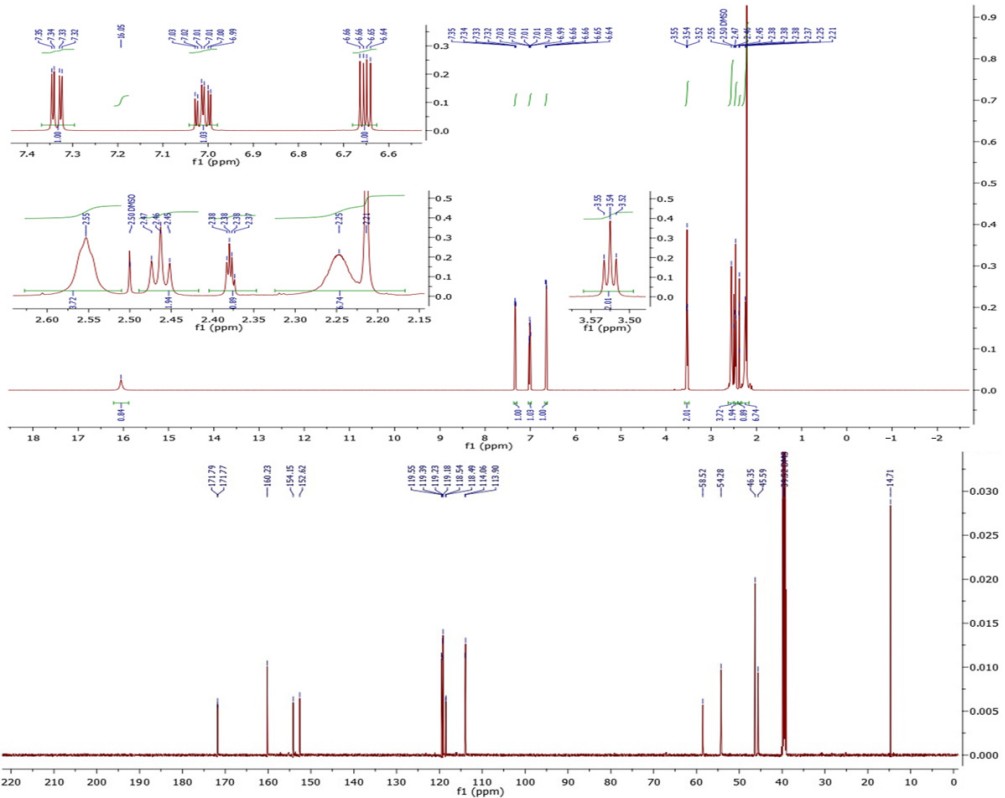

**Figure 1.** $^1$H- and $^{13}$C-NMR of the SB-functionalized ligand in DMSO-$d_6$.

The measured FT-IR transmissions of starting materials, tetradentate ligand and complex **1** are shown in Figure 2. Initially, the absence of O–H stretching band in the range of 3100–3500 cm$^{-1}$ in both 5-fluoro-2-hydroxyacetophenone starting material and its ligand could be rationalized to the robust intramolecular H-bonding [18–22] with C–OH and C=N groups. The two weak vibrational bands at 3359 and 3277 cm$^{-1}$ from piperazinyl-ethylamine group were evanesced due to the formation of SB functional moiety. Furthermore, the stretching vibrations appeared in the range of 3074 to 2845 cm$^{-1}$ were ascribed to the representative aliphatic and aromatic C–H absorption frequencies. These bands were presented as well in the copper(II) complexes. The C=N stretching band of the ligand was obvious at 1650 cm$^{-1}$, while after coordination to copper metal center it was shifted to 1621 cm$^{-1}$. This marginal shifting could be referenced to distribution of the electron density of the electron pair of nitrogen to copper metal center as well to the consequent polarization, which led to electron depopulation of the C=N-Cu moiety. Additionally, a weak broad band at 3408 cm$^{-1}$ strongly identified the coordination of a molecule of water to the copper metal center; further stretching vibrations such as C–C, C=C, C–N and C–O for other functional groups were clearly present in the spectrum.

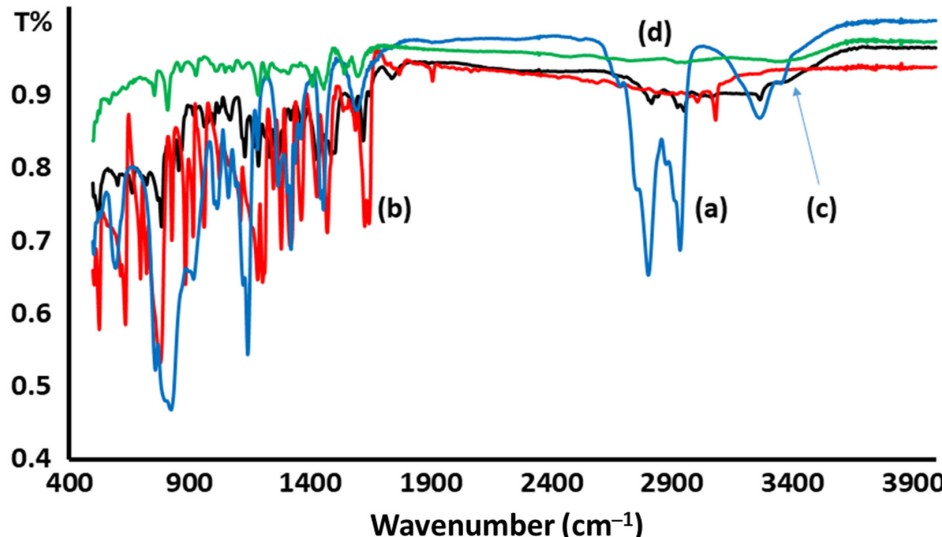

**Figure 2.** FT-IR of 2-(1-piperazinyl)ethylamine (**a**), 5-fluoro-2-hydroxyacetophenone (**b**), ligand (**c**), complex **1** (**d**).

The GC-MS spectrum of the ligand reflected the fragments with the expected molecular formula, while the CHN elemental analysis of N,N,N,O-tetradentate ligand and its complex **1–3** were fitting with the acceptable range of the proposed molecular formula and structures (Scheme 1). Likewise, Scanning Electron Microscopy (SEM) revealed the morphology of the ligand as irregular rod-shaped microstructure, whereas a micro-flower made of micro-sheets (larger than 10 μm) was matured by the Cu(II) complex, Figure 3a,b. These two materials were further analyzed via Energy Dispersive X-ray EDX. The SB-functionalized ligand reflected the presence of C, F, N and O atoms; however, additional Br and the three characteristic signals of Cu atoms were detected by complex 2 (Figure 3c,d).

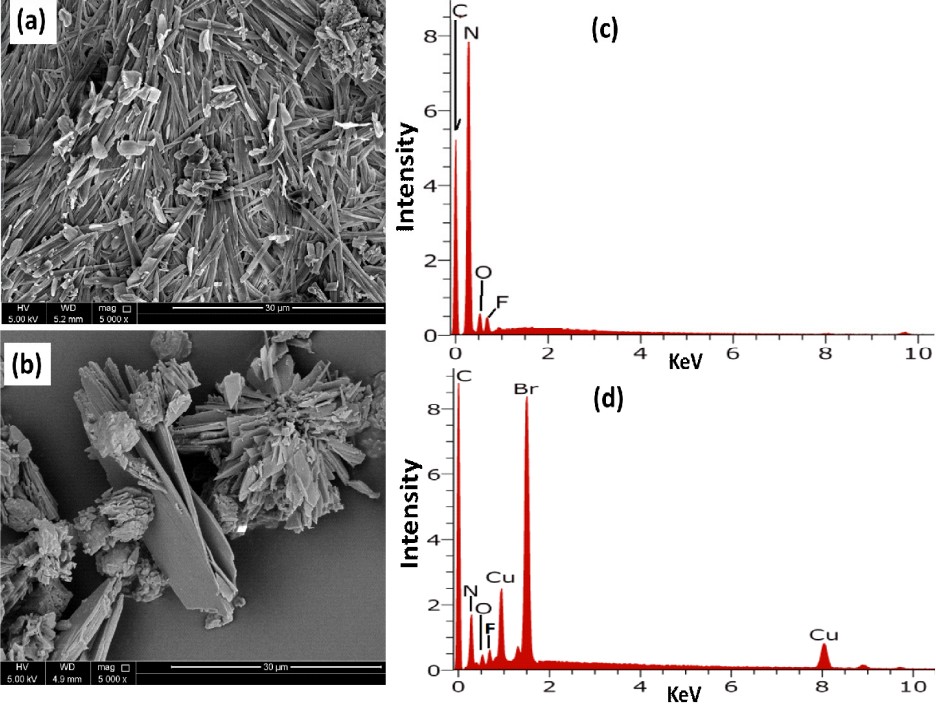

**Figure 3.** SEM images for ligand (**a**) and complex **2** (**b**); EDX for ligand (**c**) and complex **2** (**d**).

### 3.2. Electronic Absorption and Fluorescence Spectra

The UV-Vis absorption spectra of the 2-(1-piperazinyl)ethylamine, 5-fluoro-2-hydroxy-acetophenone, tetradentate ligand and complex **2** were measured in neat methanolic solution in the spectral region of 200–900 nm at ambient temperature. A maximum absorption in the UV area was recoded at 223 nm for the free amine, resulting from n-σ* electron transition, however, two broad bands were exhibited by the acetophenone derivative at $\lambda_{max}$ = 227 and 338 nm ascribed to n-σ* and π-π* electron transfer, respectively (Figure 4a,b). A broad band at $\lambda_{max}$ = 232 along with two maximum absorptions at 329 and 404 were displayed by the ligand, these electronic transitions could be assigned to n-σ*, π-π* and n-π*; the last electronic absorption designates the formation of the ligand through the condensation process, (Figure 4c). On the other hand, complex **2** exhibited two electronic transitions, a strong band with two shoulders at 240 and 265 nm, which were assigned to n-σ*, while the second wide peak at $\lambda_{max}$ = 370 nm could be resonated to ligand-to-metal charge transfer (LMCT), Figure 4d [23]. Hence, the coordination of the tetradentate ligand to copper metal center is well designated by this spectral pattern. Moreover, the d-d absorption was presented in the visible area at $\lambda_{max}$ = 675 nm, using a concentrated solution of the copper complex (Figure 4e), this outcome is in consistence with alike described complex systems [5–10].

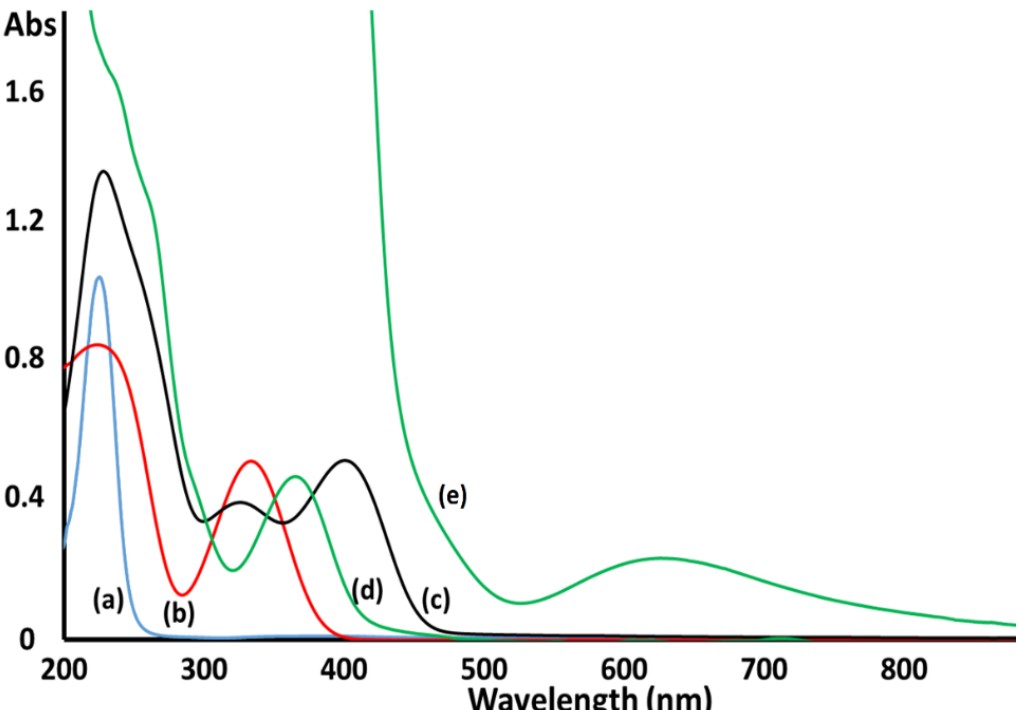

**Figure 4.** UV–Vis spectra of 2-(1-piperazinyl)ethylamine (**a**), 5-fluoro-2-hydroxyacetophenone (**b**), ligand (**c**), complex **1** (**d**), and complex **1** at a concentrated level (**e**).

The emission spectra of $1 \times 10^{-5}$ M of the tetradentate ligand and its Cu(II) complex (complex **2**) were measured at 330 nm excitation, as shown in Figure 5. It was noted that the maximum intensity of the ligand occurred at $\lambda_{max}$ = 535 nm, whereas complex **2** displayed the highest emission at $\lambda_{max}$ = 445 nm. Such a blue shift (Δλ = 90 nm) in emission is mainly resulting from ligand-Cu(II) coordination, where a typical increase in the energy gap is expected to occur as a consequence of the coordination of NNNO$^-$ to the copper metal center.

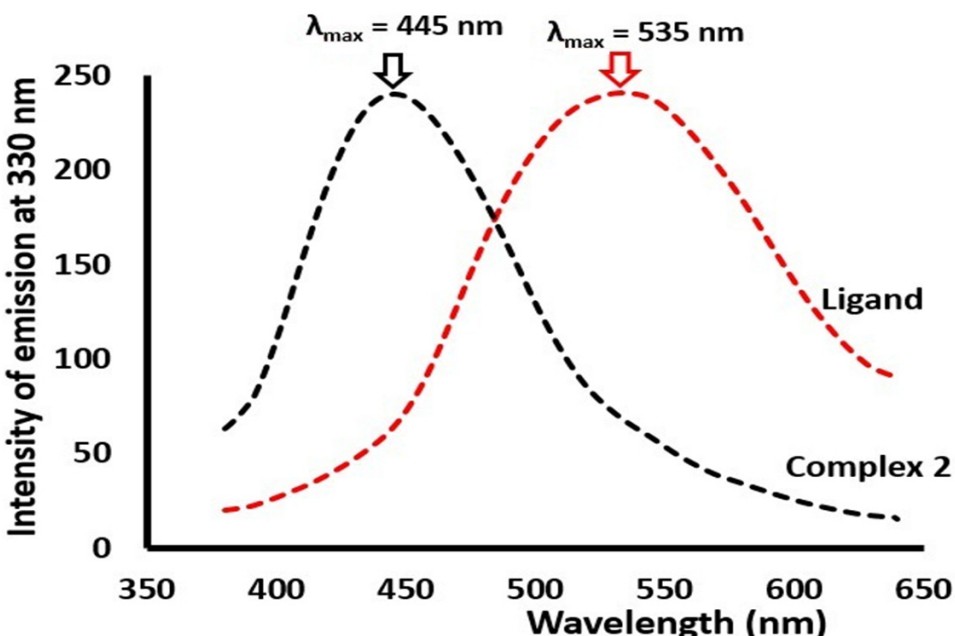

**Figure 5.** Emission spectra of $1 \times 10^{-5}$ M of ligand and complex **2** dissolved in DMSO at RT.

### 3.3. Stoichiometric Titration

Attempts to crystalize the synthesized complexes under different conditions in a suitable quality for XRD measurement were unsuccessful. Hence, the stoichiometry of complex **1** was examined using Job's method. A plot of the absorbance vs. different molar ratios of Cu(II)/ligand displayed an equimolar value of 1:1 metal:ligand as shown in Figure 6. Such stoichiometry supports the proposed octahedral Cu(II) geometry, in which the metal center is attached to a single anionic tetradentate ligand (NNNO$^-$) and a mono-halide, as presented in Scheme 1.

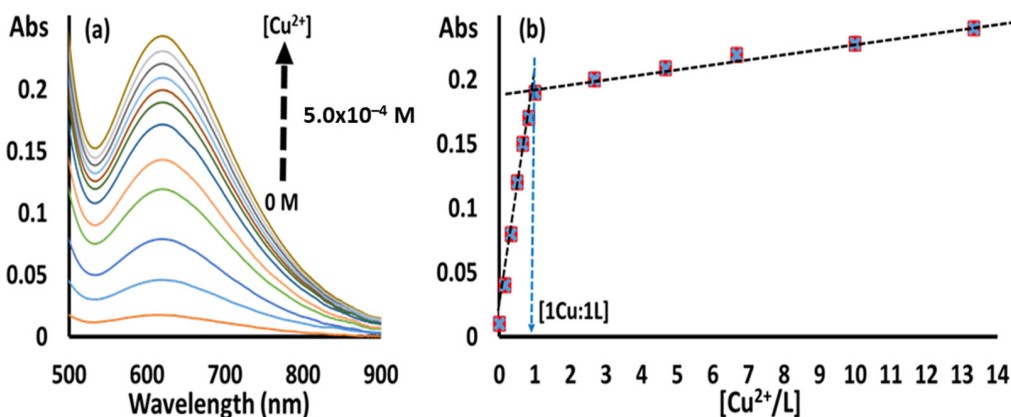

**Figure 6.** Plot of absorbance of Cu(II)-L interaction at varying concentrations of Cu(II) (**a**); Job's plot of Cu(II)-NNNO complex **1** at $\lambda_{max}$ = 620 nm (**b**).

### 3.4. Halochromism and Solvatochromism

Since Jahn-Teller effect is expected to accompany Cu(II) complexes, the color of such complexes is influenced by the change in physical and chemical conditions, making them fascinating objectives for chromotropism inspections.

The halochromism for complex 2 was investigated in pure DMF to demonstrate the reversible chromic turn off-on activity of the complex. It was observed that upon the addition of HCl, the green-colored solution was changed to colorless (turn-off) due to the

protonation process of the ligand that caused a de-structured of the complex. However, the addition of NaOH solution to the previous mixture regenerated the initial green color (turn-on) due to the de-protonation of the ligand, a process that re-structured the complex. The band intensity of d–d transition at $\lambda_{max}$ = 625 nm began to steadily decrease with increase in HCl concentration without noticeable wavelength shifting as shown in Figure 7a. Therefore, the disappearance of the green color could be ascribed to the protonation of the coordinated ligand, which causes the de-structuring process. The de-colorization process was pursued by spectrophotometric titration, revealing a consumption of mainly four equivalent protons from HCl at $\lambda_{max}$ = 625. This outcome agrees with the protonation nature of the tetradentate ligand to furnish the protonated ligand [14–26] and the hydrated copper salt ($CuBr_2$) (Figure 7b).

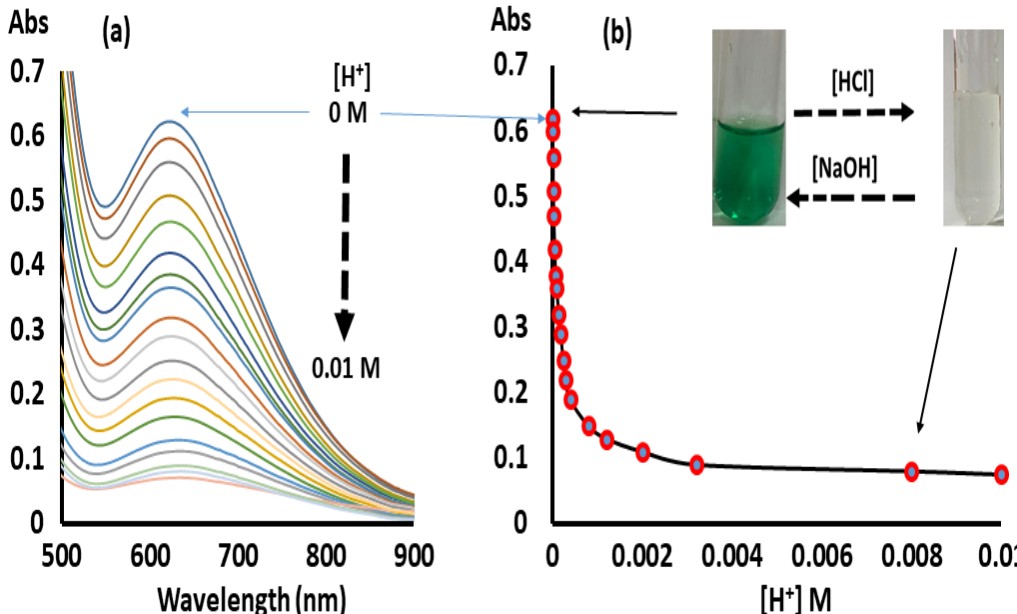

**Figure 7.** [$H^+$]-dependent visible spectra of complex **1** in water at RT (**a**) and Abs. vs. [$H^+$] relation at constant $\lambda_{max}$ = 625 nm (**b**).

Due to solubility limitation of complex **2** for the solvatochromism study, a number of solvents such as water, methanol, DMSO and DMF were investigated. Diverse colors, visible to the naked eye, were obtained by dissolving the complex in these solvents. The LMCT transition band displayed a maximum at 366 nm in water, shifted to 376 nm in methanol, 381 nm in DMF and 383 nm in DMSO (Figure 8a). Likewise, the d-d transition bands showed $\lambda_{max}$ at 618 nm in water, which shifted to 622 nm in methanol, 630 nm in DMF and 633 nm in DMSO (Figure 8b). These results indicate that both LMCT and d-d transition bands possess identical solvatochromic performance [24–30]. The experimental maxima of these two bands and Gutmann's solvent acceptor numbers (AN) [27] were set in a relation in order to explore the relative solvation. A non-linear relation was shown (a decrease followed by an increase), whereas, with Gutmann's solvent donor numbers (DN), a linear relation was revealed with correlation factors of 0.981 and 0.927 for LMCT and d-d transition bands, respectively. This behavior implies that the copper(II) complex is performing as a robust Lewis acid (Figure 8c,d).

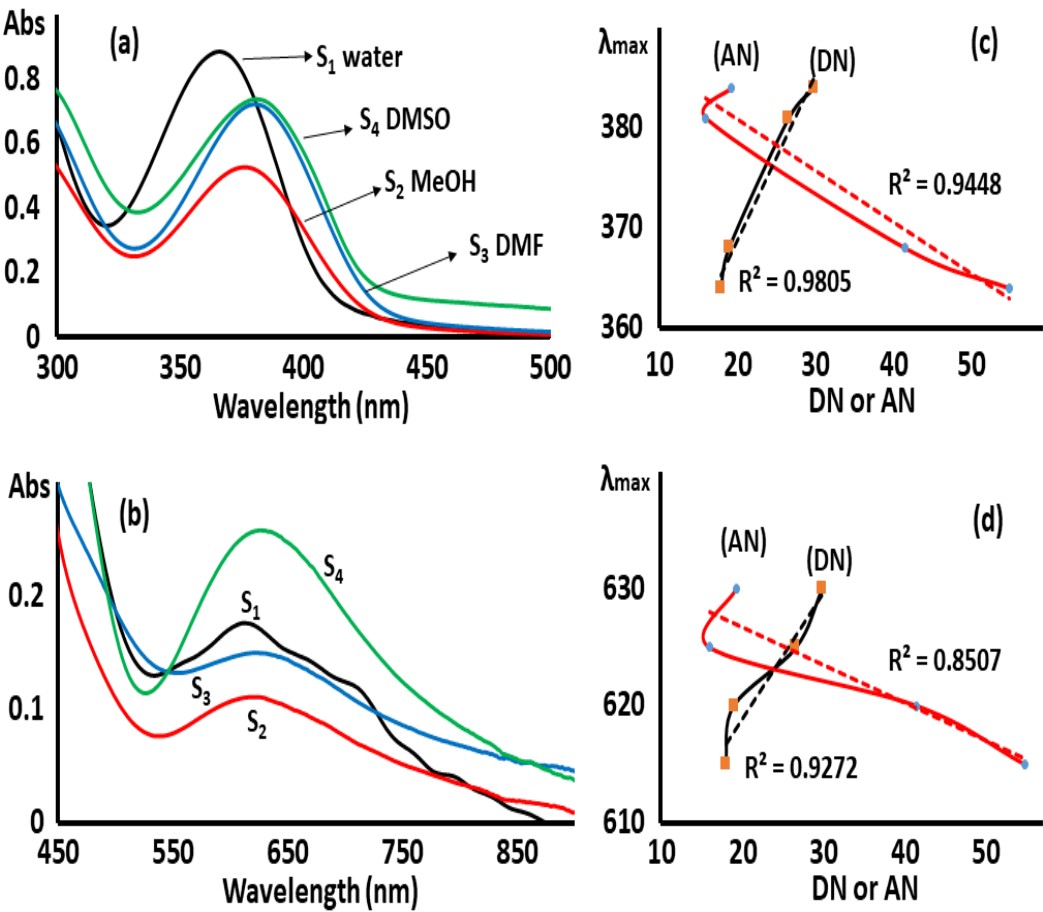

**Figure 8.** Solvatochromism of complex **2**: Abs. vs. $\lambda_{max}$ at LMCT (**a**) and at d-d (**b**), $\lambda_{max}$ vs. solvents DN/NA at LMCT (**c**) and at d-d (**d**).

### 3.5. Thermogravimetric Analysis

The thermal gravimetric analysis (TGA) and differential thermal analysis (DTA) were executed to pursue the thermal stability and level of purity of both the free tetradentate ligand and complex **2** at ambient conditions to 800 °C with a heating rate of 5 °C per minute (Figure 9). A two-step decomposition process was shown by the free ligand with no indication of possessing water or solvent molecules in its structure, reflecting an adequate stability up to 150 °C. After this degree, the compound began the first pyrolysis step until around 220 °C with 79.5% of mass loss and $T_{DTA}$ = 195 °C that could be mostly explained by losing the aryl fragment of the ligand. Subsequently, the residue was found to be stable between 220 to 415 °C before starting to completely decay at 520 °C in a broad step with $T_{DTA}$ = 450 °C, recording 19.4% mass loss. On the other hand, complex **2** displayed a simpler decomposition pattern when compared with the ligand. The first decomposition step recorded at 150–165 °C could be attributed to the loss of the $H_2O$ molecule coordinated to the metal center (Cu(L)Br·$H_2O$), measuring a mass loss of 4.3% (theoretically 4.5%) and $T_{DTA}$ = 155 °C. This dehydration step to form the Cu(L)Br complex is in consistence with the IR result. The Cu(L)Br was stable up to 550 °C before it decomposes in a single step at 550 to 640 °C with a mass loss of 77.8% (theoretically 78.2%). Both ligands (NNNO and Br) were lost from the copper metal center parallel to reaction with oxygen in one broad complex step with $T_{DTA}$ = 580 °C. Again, IR analysis confirmed the final remaining from the complex to be Cu=O with 16.9% yield (theoretically 17.2%).

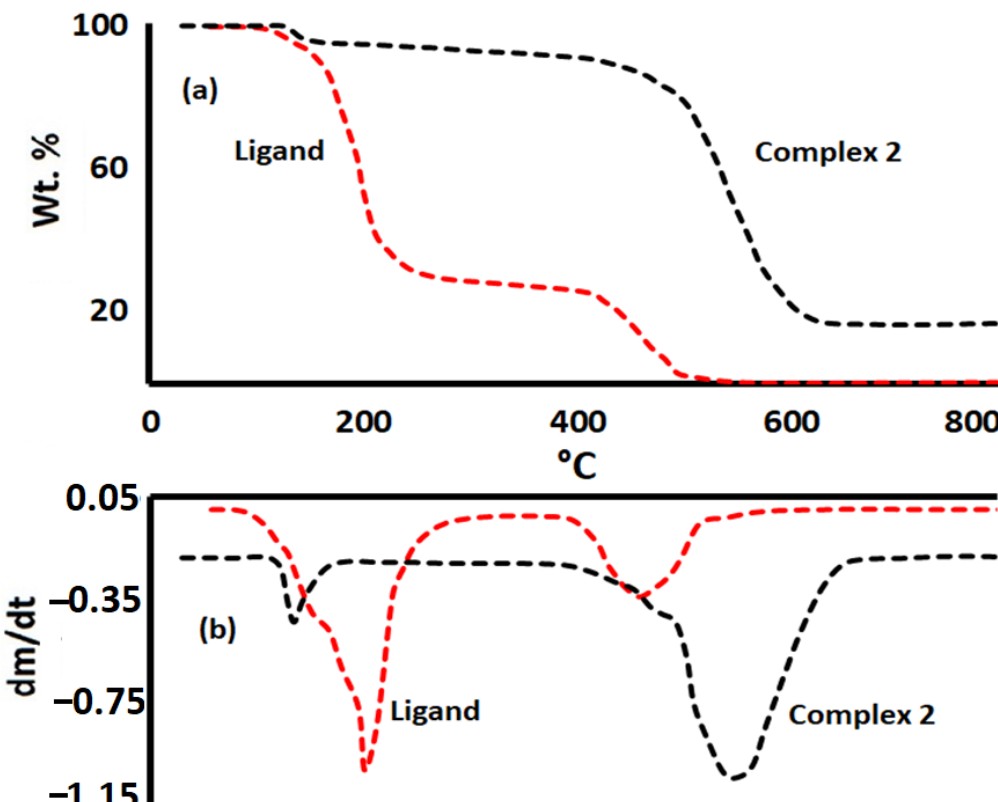

**Figure 9.** TGA (**a**) and DTG (**b**) of the tetradentate ligand and complex **2**.

### 3.6. Electrolytic Conductivity

The electrolytic conductivities of the aqueous solutions ($8 \times 10^{-5}$ M) of complexes **1–3** at 22 °C were found to be 644, 722, and 525 μS/cm, respectively. Aqueous solutions of copper precursors ($CuCl_2$, $CuBr_2$ and $Cu(NO_3)_2$) were found, as expected, to display conductivity values compared to their corresponding complexes. These observations support the expected presence of mono-halide or mono-nitrate coordinated to Cu(II) centers. Notably, the increase in temperature from 22 to 100 °C of $8 \times 10^{-5}$ M solution of complex **2** reflected an increase in the ionization percentage, which in turn increased the conductivity of the aqueous solution of the complex as exemplified in Figure 10

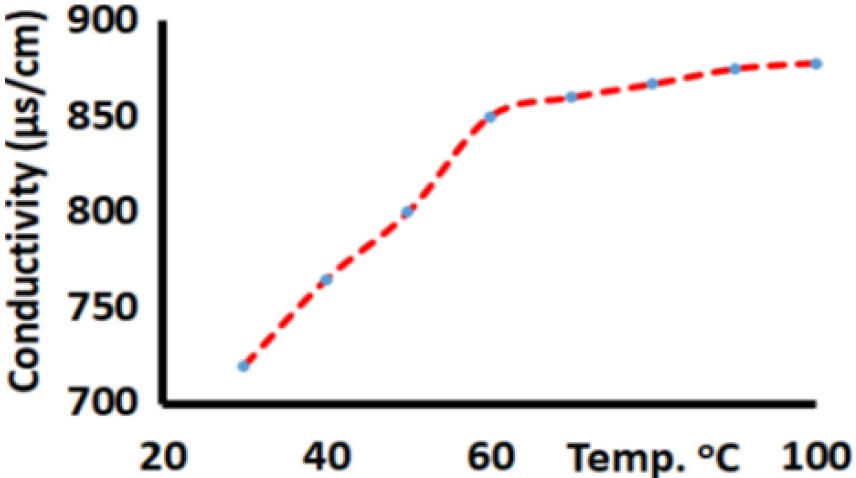

**Figure 10.** Electrolytic conductivity of complex **2**.

### 3.7. Catalytic Oxidation of Catechol

The synthesized copper complexes were examined as catalytic oxidizers for o-catechol as a model substrate to o-benzoquinone employing hydrogen peroxide as an environmentally friendly oxidant as shown in Scheme 2.

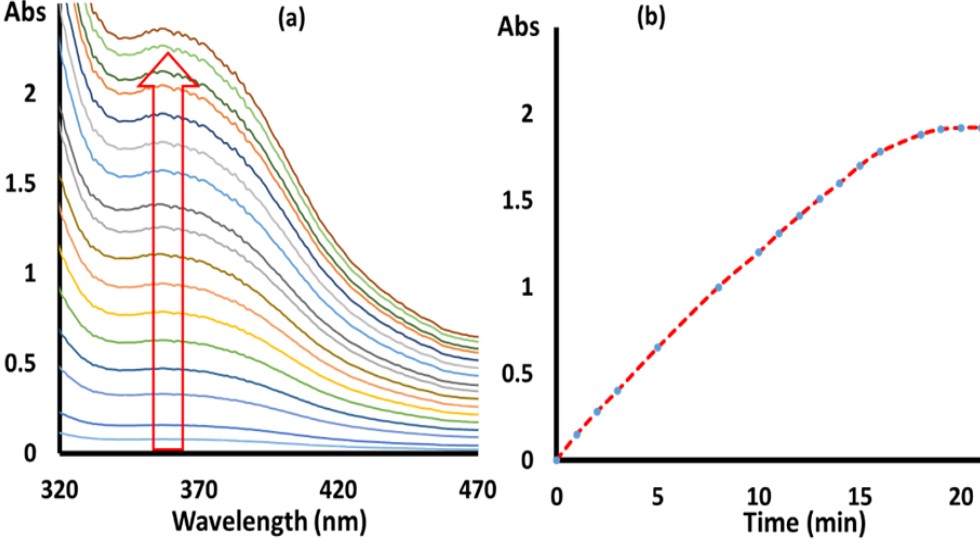

**Scheme 2.** Catalytic oxidation of o-catechol.

It is worth mentioning that plenty of sustainable catalytic oxidation reactions have been performed in the presence of molecular oxygen and hydrogen peroxide as oxidants [31]. However, peroxide is largely used for having higher oxidation potential than oxygen [31–38]. Initially the catalytic oxidation of o-catechol to o-benzoquinone was perused using UV-Vis spectroscopy, in which in the recorded absorbance values at 358 nm of the o-benzoquinone were time dependent, measured every one minute for 20 min using complex **3** (Figure 11).

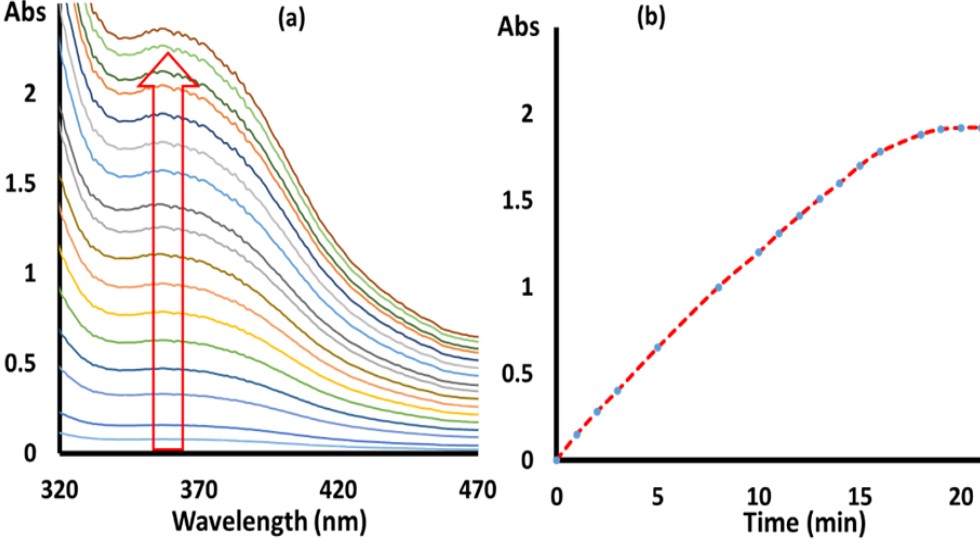

**Figure 11.** (**a**) UV-Vis spectra of formation of 1,2-benzoquinone from o-catechol using complex **3** catalytic matrix, and (**b**) relation graph between absorbance at $\lambda$ = 358 nm and time.

The conversion percentage of the staring material and reaction conditions are listed in Table 1. Generally, a mixture of 2.5 mmol of o-catechol and 5 mmol of 30% hydrogen peroxide in the presence of $2.5 \times 10^{-2}$ mmol of copper catalyst (1–3) in 10 mL of solvent was heated and stirred for certain times in a 20 mL glass tube at open atmosphere and RT, except where otherwise specified.

All copper complexes displayed an outstanding catalytic conversion of the staring material to yield the desired o-benzoquinone in water as a solvent and within very short time at room temperature (Trials 1–3). It is sensible that complex **1** demonstrated a slightly higher TOF comparing with complexes 2 and 3, that could be rationalized to the electronic effect of the coordinated chloride ion to copper metal center. Other polar solvents, such as DMSO and DMF displayed high level of conversion under the same conditions (Entries 4,5). Increasing the temperature of catalytic conversion reaction from room temperature to 50 °C allowed an increase in the conversion rate and at the same time a reduction in reaction time

by half (Trial 6), while the TOF was increased further by a factor of 10 at 70 °C in 2 min (Trial 7). In the absence of the copper catalyst (Trial 8), the reaction did not take place as only traces of the product were identified. Likewise, the formation of 1,2-benzoquinone was not accomplished when no hydrogen peroxide was utilized (Trial 9). The sensitivity and stability of the catalytic transformation was tested by adding trace amounts of HCl or NaOH solutions to the reaction mixture, revealing no product formation (Trial 10). As a result, the three complexes, particularly complex **1**, demonstrated an efficient conversion of o-benzoquinone under relatively mild reaction conditions, in water as a solvent and at a temperature of 50 °C in only 5 min. In this study, no efforts were made to study the possibility of catalyst recovery and.

Copper complexes containing Schiff bases have been applied in catalytic oxidation of various types of chemical compounds including hydrocarbons [35], cyclooctene and benzylic alcohols [36–38]. However, the conversion rates and ease of experimental conditions were not optimum when compared to the process we report in the current work, where complete conversion of o-catechol to o-benzoquinone has been achieved in a very short time and under mild reaction conditions.

## 4. Conclusions

Three neutral water-soluble copper(II) complexes bearing a novel NNNO-tetradentate Schiff base ligand were synthesized and characterized by several spectral techniques. A significant turn off-on halochromism effect was detected in the acidic and basic aqueous media, while a bathochromic solvatochromism shifts at LMCT and d-d bands were detected. The fluorescence of complex **2**, as an example, displayed a remarkable blue shift ($\Delta\lambda = 90$ nm) when compared to the spectrum of its ligand. The catalytic oxidation ability of the three complexes in aqueous medium revealed a high level of activity towards converting o-catechol to o-benzoquinone in the presence of $H_2O_2$, as an environmentally friendly oxidant, under mild reaction conditions.

**Author Contributions:** Synthesis and measurement: R.E.A.-S. and S.R.A.-M. under the supervision of I.W. and A.M.S. Characterization, measurements, manuscript preparation and editing: A.M.S., K.S.M.S. and I.W. Sample analysis, supervision and data interpretation: M.I.A. All authors have read and agreed to the published version of the manuscript.

**Funding:** This work was funded by Qatar University, grants numbers QUCG-CAS-21/22-4 and QUST-2-CAS-2019-14.

**Acknowledgments:** The authors gratefully acknowledge the funds provided by Qatar University.

**Conflicts of Interest:** The authors declare no conflict of interest.

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
