# Peer review of "Synthesis of Novel Aqua ƞ4-NNNO/Cu(II) Complexes as Rapid and Selective Oxidative Catalysts for O-Catechol: Fluorescence, Spectral, Chromotropism and Thermal Analyses"

_crystals, doi:10.3390/cryst11091072_

Round 1

Reviewer 1 Report

The revised paper improved partly.  Some descriptions of the experimental data seem to be reasonable, but there are still some insufficient points listed below.  Thus, the paper cannot be accepted for the journal of ‘crystals’.

1)     The speculated structure of boat conformation of piperazine moiety in Scheme 1 is unclear due to the absence of proof.  The evidence of the boat conformation must be indicated and discussed.  Single crystal X-ray structural analyses and EPR spectra must be performed.  As for EPR data, the hyperfine structure derived from N donor atoms should be observed.

2)     In the section of 3.6 Conductivity and magnetic susceptibility, the magnetic data are missing.  If not included, the section title must be corrected.

Author Response

Comments and Suggestions from Reviewers

Reviewer 1

The revised paper improved partly.  Some descriptions of the experimental data seem to be reasonable, but there are still some insufficient points listed below.  Thus, the paper cannot be accepted for the journal of ‘crystals’.

1)     The speculated structure of boat conformation of piperazine moiety in Scheme 1 is unclear due to the absence of proof.  The evidence of the boat conformation must be indicated and discussed.  Single crystal X-ray structural analyses and EPR spectra must be performed.  As for EPR data, the hyperfine structure derived from N donor atoms should be observed.

Thank you so much for raising up this point , Sir we have modified the Scheme of the synthesis as seen below, since we failed to obtain crystals suitable for XRD-measurement, as such complexes do not crystallize well, and based on the ref [16] where the authors are able to obtain a crystal of a very similar complex, the idea of the boat bonding by the ligand was abandoned, and the final geometry around the Cu(II) assumed to be trigonal bipyramidal [16].

Moreover, we modified this point to be:

For copper complexes similar to those reported in the current work, it has been suggested that the NNNO ligand does not coordinate via the piperazine NH part since it is already protonated by the H of the OH group [16-18]. The structure of these complexes has been proved by XRD [16]. Due to the ability of the Cu(II) center to form five-coordinate complexes, a stable trigonal bipyramidal geometry has been suggested as shown in Scheme 1. Moreover, Cu(II) center has been reported to make six bonds, where water as a moderate ligand plays a complementary role in forming six or five coordination numbers [18-25]. Notably, our ligand and its Cu(II) complexes showed high stability in addition to adequate solubility in typical polar solvents such as water, MeOH, DMSO, and DMF.

Unfortunately, we do not have the possibility for ESR measuring it is not available in the country.

2)     In the section of 3.6 Conductivity and magnetic susceptibility, the magnetic data are missing.  If not included, the section title must be corrected.

The part of magnetic susceptibility was completely removed from the manuscript and the title was corrected.

Reviewer 2 Report

The chapter 3.6 should be improved by introducing the claimed magnetic susceptibility values and explaining all the statements reffering to the conductivity measurements.

Author Response

Reviewer 2

The chapter 3.6 should be improved by introducing the claimed magnetic susceptibility values and explaining all the statements referring to the conductivity measurements.

Thank you for the note. The part of magnetic susceptibility was completely removed from the manuscript.

Thank you so much

Ismail Warad

Round 2

Reviewer 1 Report

The revised paper improved.  In this version, the descriptions of the data seem to be reasonable.  Thus, the paper can be accepted for the journal of ‘crystals’.

This manuscript is a resubmission of an earlier submission. The following is a list of the peer review reports and author responses from that submission.

Round 1

Reviewer 1 Report

The paper addresses an interesting, topical field. It refers to obtaining new compounds and their characterization is done in order to assign the structure. Also, properties that make them usable are proven.

Some observations and recommendations are still needed:

  1. The accuracy of the names
  • Rows 36 and 37 instead of “copper metal” and “metal” I recommend “copper ion”.
  • Row 37: instead of “organometallic” I recommend “ coordination compound” or ‘complexes”.
  1. Analytical data should be completed with the copper content and - at least here at Experimental part, a molecular formula should be done.
  2. Electronic spectra of the complexes in the solid state should be done in order to assign the geometry of the metal ion.
  3. Stoichiometric titration complete well the study but no offer information regarding the geometry of the complex compound.

5.      Regarding the halochromism, rows 261- 262, the authors the authors acknowledge that the complex compound decomposes and do not say what happens when the base is added. It is a halochromism. Regarding catalytic activity: the ratio substrate – catalyst considered is the opposite of the recommended ratios; it is not said if the "catalyst" has been recovered and at how many cycles it can be used.

Author Response

The paper addresses an interesting, topical field. It refers to obtaining new compounds and their characterization is done in order to assign the structure. Also, properties that make them usable are proven.

*We would like to thank you very much for reviewing our manuscript.

Some observations and recommendations are still needed:

  1. The accuracy of the names
  • Rows 36 and 37 instead of “copper metal” and “metal” I recommend “copper ion”.

*Changes were made accordingly.

  • Row 37: instead of “organometallic” I recommend “ coordination compound” or ‘complexes”.

*Thank you, the word was changed to complexes.

  1. Analytical data should be completed with the copper content and - at least here at Experimental part, a molecular formula should be done.

*Corrected accordingly

  1. Electronic spectra of the complexes in the solid state should be done in order to assign the geometry of the metal ion.

*This is a very good suggestion but, unfortunately, the solid state UV-visible is not available in our lab nor in the country, we have only solution measurements.

  1. Stoichiometric titration complete well the study but no offer information regarding the geometry of the complex compound.

*We completely agree with you, but the well-known Job’s analytical method at least can give us the metal ion to ligand ratio. Moreover, since the metal ion to ligand ratio in this work is one to one, this seen together with other spectral can help a lot in estimating the structure formula, at least, the existence of two or more ligands per one metal center was excluded.

  1. Regarding the halochromism, rows 261- 262, the authors the authors acknowledge that the complex compound decomposes and do not say what happens when the base is added. It is a halochromism. Regarding catalytic activity: the ratio substrate – catalyst considered is the opposite of the recommended ratios; it is not said if the "catalyst" has been recovered and at how many cycles it can be used.

*Regarding the halochromism, the role of acid and base have been Clearfield as:

It was observed that upon the addition of HCl, the green-colored solution was changed to colorless (turn-off) due to the protonation process of the ligand that caused a de-structured of the complex. However, the addition of NaOH solution to the previous mixture regained the initial green color (tun-on) due to the de-protonation back of the ligand that caused a re-structured of the complex.

*Regarding the ratio substrate – catalyst, thank you Sir, the procedure was corrected as:

Generally, a mixture of 2.5 mmol of o-catechol and 5 mmol of 30% hydrogen peroxide in the presence of 2.5x10-2 mmol of copper catalyst.

  • We did not try to recover the catalysts; to the text, the following sentence was inserted.

In this study, no efforts were made to study the possibility of catalyst recovery and recycling have been carried out.

Reviewer 2 Report

In this article the authors report the synthesis and spectroscopic characterization of three novel aqua ƞ4-NNNO/Cu(II) Complexes as rapid and selective oxidative catalysts for o-catechol.

After reading this manuscript, my opinion is that it has been competently done and it is technically correct. However, I do not recommend publication in crystal for the following reasons:

1) I do not think it is adequate for the readers of this journal. No crystal structures are reported. In the abstract the authors state that the most important feature of the complexes is their catalytic behavior. Therefore, a journal devoted to catalysis seems more adequate.

2) The authors used the Cu-complexes for the oxidation of catechol. There are thousands of manuscripts in the literature where Schiff base ligands are used to oxidize catechol and many of them are Cu-complexes

3) Again, if the main goal of the manuscript is the catalytic ability of the complexes, how do they compare with those already reported? This is really an important point. If these complexes are very good catalysts compared with those already reported, then the authors should demonstrate it and emphasize it to give more value to this manuscript

Author Response

In this article the authors report the synthesis and spectroscopic characterization of three novel aqua ƞ4-NNNO/Cu(II) Complexes as rapid and selective oxidative catalysts for o-catechol.

After reading this manuscript, my opinion is that it has been competently done and it is technically correct. However, I do not recommend publication in crystal for the following reasons:

1) I do not think it is adequate for the readers of this journal. No crystal structures are reported. In the abstract the authors state that the most important feature of the complexes is their catalytic behavior. Therefore, a journal devoted to catalysis seems more adequate.

Thank you very much.

We are not able to collect suitable XRD crystals from our complexes; we tried several times using several techniques without positive result.  Before we submit the manuscript this issue was discussed with the journal, Moreover, we have seen many papers published in Crystals without having XRD-solved structures. 

At the end of the introduction, we mentioned the following: Furthermore, and most importantly, the catalytic oxidation of catechol to benzoquinone was achieved in an eco-friendly process employing the synthesized complexes in the presence of H2O2 as oxidant.

This sentence sounds quite different from the reported concern.

2) The authors used the Cu-complexes for the oxidation of catechol. There are thousands of manuscripts in the literature where Schiff base ligands are used to oxidize catechol and many of them are Cu-complexes

Though we cannot find a question, we have mentioned in the manuscript the synthesis, characterization and applications of three new copper complexes. Applications include Chromotropism (Halochromism and solvatochromism) and oxidation evaluation. Thus, catalysis is one part in this manuscript.

3) Again, if the main goal of the manuscript is the catalytic ability of the complexes, how do they compare with those already reported? This is really an important point. If these complexes are very good catalysts compared with those already reported, then the authors should demonstrate it and emphasize it to give more value to this manuscript

Thank you again, the scope of the oxidation reaction was only to evaluate the efficiency of our new copper complexes. The reported outcomes in Table 1 could speak for themselves without making a comparison with the well-documented catalysts.

Reviewer 3 Report

This paper reports syntheses, chemotropism, and catalytic properties of three copper mononuclear complexes with a Schiff-base ligand.  To date, many copper complexes with Schiff-base ligands have been reported, and their functionalities and physical properties were intensively studied.  In this paper, the authors synthesized new copper complexes with an asymmetric tetradentate Schiff-base ligand with a piperazine group.  These complexes were characterized by elemental analyses, EDX data, IR spectra, and thermogravimetric analyses.  Unfortunately, the molecular structure of the synthesized compounds hasn’t been determined by single-crystal X-ray crystallography.  The catalytic behavior of the compounds for the oxidation reaction of benzyl alcohol has been investigated.  The descriptions of the experimental data seem to be reasonable, but there are some insufficient points listed below.  Thus, the paper can be accepted for the journal of ‘crystals’ after major revision.

1)     Several copper complexes with Schiff-base ligands including piperazine moieties have been reported.  According to the molecular structures of similar complexes, the piperazine group forms a chair conformation and the outer nitrogen atom doesn’t coordinate to copper ion.  The authors describe boat conformation of piperazine moiety in Scheme 1.  The evidence of the boat conformation must be indicated and discussed.  Please check the following literature.

New Journal of Chemistry (2017), 41(22), 13625-13646.

RSC Advances (2015), 5(100), 82179-82191.

2)     The piperazine group can be protonated, affording piperazinium cation.  In the case of the titration experiment of HCl, the authors didn’t mention the reason for the color changes in detail.  Please discuss the species in solution during halochromism and solvatochromism. 

3)     In the section on magnetic properties, the obtained values (130, 125, and 110 for complexes 1-3, respectively) must be explained.  The unit should be shown. 

Author Response

1)     Several copper complexes with Schiff-base ligands including piperazine moieties have been reported.  According to the molecular structures of similar complexes, the piperazine group forms a chair conformation and the outer nitrogen atom doesn’t coordinate to copper ion.  The authors describe boat conformation of piperazine moiety in Scheme 1.  The evidence of the boat conformation must be indicated and discussed.  Please check the following literature.

We have discussed clearly this point using the suggested references.

It has been suggested that the piperazine part of the NNNO ligand coordinated via both of its N atoms resulting the boat [16] and not chair conformation [17, 18] in response to the desire of the Cu(II) center of having six coordination number, thus reach the stable octahedron geometry. Moreover, it was noted that Cu(II) center desires also the possibility of bonding with five bonds, where water as moderate ligand plays the complementary role in having six or five coordination numbers as seen in Scheme 1.

[16] S. Kumari, A. K. Mahato, A. Maurya, V. K.Singh, N. Keshwarwani, P. Kachhap, I. O. Koshevoy, C, Haldar, New J. Chem., 41 (2017) 13625.

[17] T. Maity, D. Saha, S. Bhunia, P. Brandão, S. Das, S. Koner, RSC Advances 5 (2015), 8217.

[18] K. S. M. Salih, A. M. Shraim, S. R. Al-Mhini, R. E. Al-Soufi, I. Warad, Emergent Materials 4 (2021) 423.

2)     The piperazine group can be protonated, affording piperazinium cation.  In the case of the titration experiment of HCl, the authors didn’t mention the reason for the color changes in detail.  Please discuss the species in solution during halochromism and solvatochromism. 

The halochromism for complex 2 was investigated in pure DMF to demonstrate the reversible chromic turn off-on activity of the complex. It was observed that upon the addition of HCl, the green-colored solution was changed to colorless (turn-off) due to the protonation process of the ligand that caused a de-structured complex. However, the addition of NaOH solution to the previous mixture regained the initial green color (tun-on) due to the de-protonation back of the ligand that caused a re-structured of the complex. The band intensity of d–d transition at λmax = 625 nm began to steadily decrease with an increase in HCl concentration without noticeable wavelength shifting as shown in Figure 7a. Therefore, the disappearance of the green color could be ascribed to the protonation of the coordinated ligand, which causes the de-structuring process.

3)     In the section on magnetic properties, the obtained values (130, 125, and 110 for complexes 1-3, respectively) must be explained.  The unit should be shown.

Thank you sir for this good point, the magnetic susceptibility values were taken directly from the machine at constant using the same cell volume, conc, temperature, etc. therefore, the values in unitless. Now the magnetic susceptibility values were corrected to cm3/mol unit, therefore it becomes 0.26, 0.25, and 0.22 respectively for complexes 1-3. Moreover, figure 10b was also corrected and inserted into the text.

Round 2

Reviewer 2 Report

My opinion regarding this manuscript has not changed. The authors have not considered my previous suggestions when preparing the revised version. I do not think it deserves publication for two reasons:

1) The topic is not adequate for the readeship of this journal

2) The reaction is not new, in fact it has been studied by a great deal of manuscripts previously and the authors have not compared the catalytic efficiency of their Cu-complexes with other catalysts.

Reviewer 3 Report

The paper has been revised according to the referees’ comments.  The descriptions of the experimental data have been changed, but there are some insufficient points listed below.  Thus, the paper cannot be accepted for the journal of ‘crystals’ in this stage.

1)     The authors must show clear evidence of sin-coordinate copper ion with NNNO-tetradentate coordination fashion.  EPR measurement for the complex will provide the hyperfine structure of N donor atoms.  It’s better to perform single-crystal X-ray structural analyses.  XAFS can give some information on coordination environments.

2)     To discuss the halochromism, the colorless solution after HCl titration must be analyzed.  How about ESI-MS spectra?  This color change is not surprising behavior.  The authors must emphasize the advantage of the present compounds.

3)           The descriptions of magnetic susceptibilities are wrong and strange.  XmT values must be discussed comparing with spin-only values.  The XmT value of Cu(II) ion (S = 1/2) at room temperature should be close to 0.375 emu mol-1 K.